# Utility of TEMPS-A in differentiation between major depressive disorder, bipolar I disorder, and bipolar II disorder

Chihiro Morishita[1], Rie Kameyama[2], Hiroyuki Toda[3], Jiro Masuya[1], Masahiko Ichiki[1], Ichiro Kusumi[4], Takeshi Inoue[1]*

1 Department of Psychiatry, Tokyo Medical University, Shinjuku-ku, Tokyo, Japan, 2 Department of Neuropsychiatry, Takikawa Municipal Hospital, Takikawa-shi, Hokkaido, Japan, 3 Department of Psychiatry, National Defense Medical College, Tokorozawa-shi, Saitama, Japan, 4 Department of Psychiatry, Hokkaido University Graduate School of Medicine, Sapporo-shi, Hokkaido, Japan

* tinoue@tokyo-med.ac.jp

## Abstract

**Data Availability Statement:** Data cannot be shared publicly because of Ethics Committee restriction. All relevant data are within the paper. Data are available from the Internal Review Board of the Department of Psychiatry, Tokyo Medical

### Background

The association between temperament characteristics and mood disorders has gained much attention in recent years. The Temperament Evaluation of Memphis, Pisa, Paris and San Diego-autoquestionnaire version (TEMPS-A) is a self-rating scale measuring 5 affective temperament dimensions. In this study, we aimed to clarify whether each affective temperament of TEMPS-A is a differentiating factor between major depressive disorder (MDD), bipolar I disorder (BD-I), and bipolar II disorder (BD-II), and analyzed the utility of TEMPS-A in their differential diagnosis in a clinical setting.

### Methods

A total of 346 patients (MDD, n = 176; BD-II, n = 112; BD-I, n = 58) filled out TEMPS-A. To assess the patients' mood state at the time of temperament assessment, Patient Health Questionnaire-9 (PHQ-9) and Young Mania Rating Scale (YMRS) were also conducted.

### Results

Multivariate logistic regression analysis demonstrated that cyclothymic and anxious temperament scores were significant factors differentiating the diagnosis of BD-I and BD-II from the diagnosis of MDD, and hyperthymic temperament score was a specific factor for the differential diagnosis of BD-I versus the diagnosis of BD-II.

### Limitations

All of the patients included in our study received treatment in large general hospitals. Because the nature of the present study was cross-sectional, some MDD subjects in this study might have unrecognized BD-I/BD-II.

University (Japan)(contact via email: seisinka@tokyo-med.ac.jp) for researchers who meet the criteria for access to confidential data.

**Funding:** This work was supported in part by a Grant-in-Aid for Scientific Research (no. 16K10194, to T. Inoue) from the Japanese Ministry of Education, Culture, Sports, Science and Technology, Research and Development Grants for Comprehensive Research for Persons with Disabilities from Japan Agency for Medical Research and Development, and a grant from SENSHIN Medical Research Foundation.

**Competing interests:** I have read the journal's policy and the authors of this manuscript have the following competing interests: Jiro Masuya has received personal compensation from Otsuka Pharmaceutical, Eli Lilly, Astellas, and Meiji Yasuda Mental Health Foundation, and grants from Pfizer. Masahiko Ichiki has received personal compensation from Otsuka Pharmaceutical, Pfizer, Eli Lilly, Mitsubishi Tanabe Pharma, Mochida Pharmaceutical, Meiji Seika Pharma, Janssen Pharmaceutical, Takeda Pharmaceutical, MSD, Dainippon Sumitomo Pharma, and Eisai; grants from Otsuka Pharmaceutical, Eli Lilly, Eisai, Shionogi, Takeda Pharmaceutical, MSD, and Pfizer; and is a member of the advisory board of Meiji Seika Pharma. Ichiro Kusumi has received personal compensation from Astellas, Chugai Pharmaceutical, Daiichi Sankyo, Dainippon Sumitomo Pharma, Eisai, Eli Lilly, Janssen Pharmaceutical, Kyowa Hakko Kirin, Meiji Seika Pharma, MSD, Nippon Chemiphar, Novartis Pharma, Ono Pharmaceutical, Otsuka Pharmaceutical, Pfizer, Mitsubishi Tanabe Pharma, Shionogi, and Yoshitomiyakuhin; has received research/grant support from AbbVie GK, Asahi Kasei Pharma, Astellas, Boehringer Ingelheim, Chugai Pharmaceutical, Daiichi Sankyo, Dainippon Sumitomo Pharma, Eisai, Eli Lilly, GlaxoSmithKline, Kyowa Hakko Kirin, Meiji Seika Pharma, MSD, Novartis Pharma, Ono Pharmaceutical, Otsuka Pharmaceutical, Pfizer, Takeda Pharmaceutical, Tanabe Mitsubishi Pharma, Shionogi, and Yoshitomiyakuhin; and is a member of the advisory board of Dainippon Sumitomo Pharma and Tanabe Mitsubishi Pharma. Takeshi Inoue has received personal compensation from Mochida Pharmaceutical, Takeda Pharmaceutical, Eli Lilly, Janssen Pharmaceutical, MSD, Taisho Toyama Pharmaceutical, Yoshitomiyakuhin, and Daiichi Sankyo; grants from Shionogi, Astellas, Tsumura, and Eisai; grants and personal fees from Otsuka Pharmaceutical, Dainippon Sumitomo Pharma, Mitsubishi Tanabe Pharma, Kyowa Pharmaceutical Industry, Pfizer, Novartis Pharma, and Meiji Seika

## Conclusions

Cyclothymic and anxious temperament scores assessed by TEMPS-A might enable differentiation between MDD and BD, and hyperthymic temperament score on TEMPS-A might be useful in distinguishing between BD-I and BD-II.

## Introduction

Differentiating between major depressive disorder (MDD), bipolar I disorder (BD-I), and bipolar II disorder (BD-II) in the early stages of disease is clinically important [1, 2], because clinicians should take different treatment approaches for the 3 disorders and inappropriate treatment can be associated with poor prognoses. For example, clinicians should not provide antidepressant monotherapy to patients with BD, particularly those with BD-I, according to many treatment guidelines for mood disorders [3, 4], and inappropriate treatment of BD and an extended duration of untreated BD may increase the risk of mood instabilities and suicide attempts [1, 2]. However, this distinction often requires a large amount of time and effort for clinicians, owing to various reasons. One reason is that in two-thirds of patients, the onset of BD is a major depressive episode [5]. Another reason is the lack of patient self-awareness of prior or future manic or hypomanic episodes [6].

In recent years, the association between mood disorders and temperament characteristics has gathered much attention. The Temperament Evaluation of Memphis, Pisa, Paris and San Diego-autoquestionnaire version (TEMPS-A) is a 110-item true-false self-reported questionnaire that quantitatively assesses 5 domains of affective temperament, i.e., depressive, hyperthymic, cyclothymic, irritable, and anxious temperaments [7]. Vazquez et al. showed the association between some affective temperaments on TEMPS-A and a suicidal risk in both psychiatric and general population samples [8]. The association between some affective temperaments on TEMPS-A and treatment resistance in MDD and BD patients has also been studied [9–11]. Moreover, Goto et al. suggested that cyclothymic and hyperthymic temperaments were associated with bipolarity, and also investigated the association between treatments and remission rates in patients with bipolarity [12]. Furthermore, several studies have shown the possibility of the usefulness of TEMPS-A to differentiate between MDD and BD [13–18]. However, most previous studies had limitations, such as a small sample size or that the data were not analyzed by multivariate analyses, including current mental status as an independent variable, although mood state is known to influence the self-evaluation of affective temperament [19]. Moreover, Solmi et al. [20] performed a meta-analysis and found that patients with a diagnosis of BD had significantly higher cyclothymic, hyperthymic, and irritable temperament scores compared with patients with a diagnosis of MDD. To our knowledge, this is the only study to date that performed a meta-analysis to assess the association between affective temperament scores on TEMPS-A and the diagnosis of mood disorders; however, the subjects included were of various mental states, and the possible effects of their mental states were ignored in the meta-analysis.

On the other hand, our previous studies demonstrated that cyclothymic and anxious temperaments were factors differentiating MDD and BD patients from healthy subjects [9, 10]. The data of our studies were analyzed by multivariate logistic regression analyses to take into account several relevant factors, such as depression severity, but we did not include manic or hypomanic symptoms in the analyses. Moreover, we did not compare the characteristics of TEMPS-A between MDD patients and BD patients, or between BD-I patients and BD-II

Pharma; and is a member of the advisory boards of Pfizer, Novartis Pharma, and Mitsubishi Tanabe Pharma. All other authors declare that they have no actual or potential conflicts of interest associated with this study. This does not alter our adherence to PLOS ONE policies on sharing data and materials.

patients. Takeshima and Oka [11] demonstrated that cyclothymic and hyperthymic temperaments are independent differentiating factors of BD in their comparison between MDD and BD patients. The data of their study were analyzed by multivariate logistic regression analysis, including the severity of depressive symptoms as an independent variable, but they ignored manic symptoms at the time of temperament assessment, and they did not consider BD-I and BD-II separately.

There have been few studies in which the severity of mood symptoms and the effects of other temperaments were taken into account in the comparison between MDD and BD. In other words, in previous studies, the effects of relevant factors were not considered. Moreover, no study to date has performed multivariate analysis to identify affective temperaments as differentiating factors of BD-I compared with BD-II. For these reasons, it has not yet been clarified whether TEMPS-A is useful for the differential diagnosis of mood disorders. Therefore, we hypothesized that each subscale of affective temperaments assessed by TEMPS-A is a differentiating factor of the diagnosis of MDD, BD-I, and BD-II.

The aim of this study was to test this hypothesis, and to clarify whether TEMPS-A is useful for a differential diagnosis in a clinical setting.

## Subjects and methods

### Subjects

Subjects were 157 outpatients and 189 inpatients (MDD, n = 176; BD-II, n = 112; BD-I, n = 58). Approximately 70 or more subjects for each disorder group were required for our analyses, because we planned to conduct multivariate logistic regression analyses including 7 factors as independent variables. Patient diagnoses were performed by psychiatric specialists who were responsible for the treatment of these patients, according to the Diagnostic and Statistical Manual of Mental Disorders, Fourth Edition, Text Revision (DSM-IV-TR). The outpatients were those who received treatment at the Department of Psychiatry of either Hokkaido University Hospital or Self-Defense Forces Sapporo Hospital, both in Sapporo, National Defense Medical College Hospital in Tokorozawa, or Self-Defense Forces Central Hospital in Tokyo, between April 2012 and April 2013. The inpatients were those who received treatment at the Department of Psychiatry, Hokkaido University Hospital, between January 2010 and December 2017. The inclusion criteria of the patients were as follows: (1) a principal diagnosis of MDD, BD-I, or BD-II according to the DSM-IV-TR criteria; (2) 20 years of age or older; (3) the ability to complete the self-reported questionnaires; and (4) the ability to provide written informed consent. The exclusion criteria were as follows: (1) having serious physical or mental symptoms that hinder the completion of the self-reported questionnaires; (2) having organic mental disorders or a previous history of them; (3) meeting the diagnostic criteria of substance-use disorders; and (4) having a diagnosis of axis II according to the DSM-IV-TR criteria. Patients meeting the eligibility criteria were informed about our research by their doctors in charge, and those who gave written consent were included in the study. The sample might not be a nationally representative sample, as the sample was limited to a convenience sample, and consisted of only patients who received treatment at general or university hospitals in specific parts of Japan. In other words, the sample did not include patients who received treatment at psychiatric hospitals or clinics. The present study was performed in accordance with the Declaration of Helsinki, and was approved by the ethics committees of National Defense Medical College, Hokkaido University Hospital, and Tokyo Medical University. The approval number is SH4098.

Table 1 presents the demographic data, Patient Health Questionnaire-9 (PHQ-9) and Young Mania Rating Scale (YMRS) scores, and TEMPS-A subscores of the subjects.

**Table 1. Demographic and clinical characteristics and TEMPS-A scores of the patients analyzed in this study.**

| | MDD (n = 176) | BD-II (n = 112) | BD-I (n = 58) | Statistical difference | |
|---|---|---|---|---|---|
| Demographics | | | | | |
| Age, years: mean (S.D.) | 46.5 (13.1) | 45.5 (12.7) | 49.2 (11.5) | $F(2,343) = 1.66$ | $p = 0.193$ |
| Sex (male): n (%) | 92 (52.3) | 49 (43.8) | 35 (60.3) | | $p = 0.106$ |
| Education, years: mean (S.D.) | 14.0 (2.4) | 14.2 (2.4) | 14.7 (1.9) | $F(2,343) = 2.03$ | $p = 0.133$ |
| Employment status (employed): n (%) | 64 (36.4) | 38 (33.9) | 14 (24.1) | | $p = 0.764$ |
| Marital status (married): n (%) | 91 (51.7) | 63 (56.3) | 32 (55.2) | | $p = 0.237$ |
| Clinical features | | | | | |
| PHQ-9 score: mean (S.D.) | 8.9 (7.0) | 9.0 (6.1) | 8.5 (7.6) | $F(2,343) = 0.13$ | $p = 0.881$ |
| YMRS score: mean (S.D.) | 0.7 (2.2) | 1.6 (2.8)* | 1.8 (2.9)* | $F(2,343) = 5.92$ | $p = 0.003$ |
| TEMPS-A scores | | | | | |
| Depressive score: mean (S.D.) | 1.50 (0.21) | 1.51 (0.22) | 1.49 (0.17) | $F(2,343) = 0.22$ | $p = 0.800$ |
| Cyclothymic score: mean (S.D.) | 1.29 (0.22) | 1.40 (0.27)** | 1.43 (0.30)** | $F(2,343) = 10.75$ | $p = 0.000$ |
| Hyperthymic score: mean (S.D.) | 1.19 (0.17) | 1.19 (0.16) | 1.27 (0.25) **# | $F(2,343) = 5.15$ | $p = 0.006$ |
| Irritability score: mean (S.D.) | 1.19 (0.16) | 1.23 (0.20) | 1.21 (0.22) | $F(2,343) = 1.79$ | $p = 0.168$ |
| Anxiety score: mean (S.D.) | 1.42 (0.25) | 1.42 (0.24) | 1.41 (0.26) | $F(2,343) = 0.56$ | $p = 0.946$ |

MDD, major depressive disorder; BD-I, bipolar I disorder; BD-II, bipolar II disorder; S.D., standard deviation; PHQ-9, Patient Health Questionnaire-9; YMRS, Young Mania Rating Scale; TEMPS-A, Temperament Evaluation of Memphis, Pisa, Paris and San Diego-autoquestionnaire

*$p < 0.05$ vs MDD

**$p < 0.01$ vs MDD

#$p < 0.05$ vs BD-II.

## Measures and procedures

The patients completed the Japanese standardized version of the TEMPS-A. The validity and reliability of TEMPS-A for psychiatric disorders, particularly for mood disorders, was suggested by Akiskal et al. [7], and the validity and reliability of the Japanese version was also shown by Matsumoto et al. [14]. In the analyses, True was scored as 2 and False was scored as 1, and final values were obtained by dividing the total points of each affective temperament subscale by the number of question items measuring each subscale.

The inpatients filled out the TEMPS-A at the time of hospital discharge, after their symptoms were improved by adequate treatment. The outpatients filled out the TEMPS-A at some point in their continuous visits, but not at their first visit. In other words, they completed it after they received treatment for months or years.

To assess the severity of depressive symptoms, patients were also asked to complete the Japanese version of PHQ-9 at the same time as completing the TEMPS-A. The PHQ-9 is a self-reported questionnaire consisting of 9 items on a 4-point Likert scale, used as an index of depressive symptom severity and as a screening test for major depressive episodes. The Japanese version of the PHQ-9 was used in this study. Some studies suggested that the PHQ-9 yields an index of depressive symptom severity and has diagnostic validity, and also validates the Japanese version of the PHQ-9 [21–23]. Additionally, to assess manic symptoms, evaluation using the Young Mania Rating Scale (YMRS) was performed by psychiatrists. The YMRS is an assessment sheet consisting of 11 items, which are each rated on a 0–4 scale. The rating of severity is based on both the subjective statements of patients and objective observation by clinicians. The construct validity and reliability of YMRS was demonstrated by Young et al. [24]. Demographic features, such as age, sex, duration of education, employment, and marital status were also analyzed. Data were anonymized and sent to our research group.

## Statistical analyses

Continuous variables were compared using analysis of variance, and categorical variables were compared by the Kruskal-Wallis test. Post-hoc analyses were performed by the Bonferroni test. Multivariate logistic regression analyses using the backward stepwise method were performed to identify factors of the distinction of MDD, BD-I, and BD-II. Then, receiver operating characteristic (ROC) curves were used to assess the performances. Moreover, multivariate logistic regression analyses using the forced entry method were performed to confirm the reproducibility of the results.

Statistical analyses were performed using SPSS 24.0J for Windows (SPSS Inc., Chicago, IL, USA). A *p*-value of less than 0.05 was considered to indicate a statistically significant difference.

# Results

## Demographic and clinical characteristics and TEMPS-A subscale scores (Table 1)

YMRS scores were higher in the BD-I and BD-II groups than in the MDD group, although a statistically significant difference between the BD-I and BD-II groups was not detected. Differences of age, sex, education years, employment status, marital status, and PHQ-9 scores were not statistically significant among the 3 disease groups.

Three-group comparisons of MDD, BD-I, and BD-II demonstrated statistically significant differences in cyclothymic and hyperthymic temperaments, but we did not find statistically significant differences in depressive, irritable, or anxious temperaments among the 3 groups. Substantially higher cyclothymic temperament scores were found in the BD-I and BD-II groups than in the MDD group, and substantially higher hyperthymic temperament scores were found in the BD-I group than in the MDD and BD-II groups.

## Analysis of differentiating factors of the diagnosis of MDD, BD-I, and BD-II

Taking into account the results shown in Table 1, multivariate logistic regression analysis was conducted for the severity of mood symptoms and TEMPS-A subscores, to identify independent differentiating factors of the diagnosis of mood disorders.

In Table 2, we summarized the results of backward stepwise multivariate logistic regression analysis performed to identify independent differentiating factors of the diagnosis of BD-I versus the diagnosis of MDD. Among the PHQ-9 scores, YMRS scores, and 5 TEMPS-A subscale scores, cyclothymic and anxious temperament scores on the TEMPS-A as well as YMRS scores were significant independent differentiating factors of the diagnosis of BD-I versus the diagnosis of MDD, whereas PHQ-9 scores and depressive, hyperthymic, and irritable temperament scores on the TEMPS-A were not considered as significant differentiating factors of the diagnosis of BD-I.

Table 3 demonstrates the results of stepwise multivariate logistic regression analysis conducted to identify variables distinguishing BD-II from MDD. Cyclothymic and anxious temperament scores were identified as significant independent differentiating factors of the diagnosis of BD-II versus the diagnosis of MDD, whereas the other variables were not considered as significant differentiating factors of the diagnosis of BD-II.

As shown in Table 4, stepwise multivariate logistic regression analysis of the diagnosis of BD-I compared with the diagnosis of BD-II demonstrated that only hyperthymic temperament

**Table 2. Stepwise multivariate logistic regression analysis of the diagnosis of MDD and BD-I.**

| Variable | Stepwise analysis | | | | |
|---|---|---|---|---|---|
| | B | S.E. | *p*-value | OR | 95% CI |
| Cyclothymic temperament | 3.68 | 0.87 | 0.000 | 39.52 | 7.14–218.75 |
| Anxious temperament | −2.44 | 0.88 | 0.005 | 0.09 | 0.02–0.49 |
| YMRS score | 0.13 | 0.06 | 0.049 | 1.13 | 1.00–1.28 |
| Constant | −2.79 | 0.99 | 0.005 | 0.06 | |

Fit index of this model: $\chi^2$ = 28.11 (*p*-value < 0.05), Cox-Snell $R^2$ = 0.11, Hosmer–Lemeshow test *p* = 0.275, sensitivity = 0.17, specificity = 0.96, positive predictive value = 0.56, negative predictive value = 0.78, AUC of ROC = 0.71

dependent variable: the diagnosis of MDD (1) and BD-I (2)

7 independent variables: scores of 5 subscales of the TEMPS-A and the severity of depressive and manic symptoms (PHQ-9 and YMRS scores, respectively).

AUC, area under the curve; BD- I, bipolar I disorder; CI, confidence interval; MDD, major depressive disorder; OR, odds ratio; PHQ-9, Patient Health Questionnaire-9; ROC, receiver-operating characteristic; B, partial regression coefficient; S.E., standard error; TEMPS-A, Temperament Evaluation of Memphis, Pisa, Paris and San Diego-autoquestionnaire; YMRS, Young Mania Rating Scale.

score was a significant independent differentiating factor correlating with the diagnosis of BD-I.

We also conducted the multivariate logistic regression analyses using the forced entry method, including PHQ-9 and YMRS scores, and TEMPS-A subscores as independent variables, to confirm the robustness of our results. The results are shown in S1–S3 Tables. S1 and S2 Tables show that only cyclothymic temperament score is a significant differentiating factor of the diagnosis of BD-I and BD-II from the diagnosis of MDD. S3 Table demonstrates that hyperthymic temperament score is a specific factor for the differential diagnosis of BD-I versus the diagnosis of BD-II.

## Discussion

The principle findings of our study are as follows. Cyclothymic temperament and anxious temperament may differentiate the diagnosis of BD-I and BD-II compared with MDD, and hyperthymic temperament may differentiate the diagnosis of BD-I compared with BD-II. These findings supported the hypothesis that some affective temperaments assessed by TEMPS-A can be used as differentiating factors of the diagnosis of MDD, BD-I, and BD-II. Affective temperament is a concept proposed by Akiskal and his colleagues. They described

**Table 3. Stepwise multivariate logistic regression analysis of the diagnosis of MDD and BD-II.**

| Variable | Stepwise analysis | | | | |
|---|---|---|---|---|---|
| | B | S.E. | *p*-value | OR | 95%CI |
| Cyclothymic temperament | 2.91 | 0.72 | 0.000 | 18.32 | 4.44–75.65 |
| Hyperthymic temperament | −0.73 | 0.80 | 0.363 | 0.48 | 0.10–2.32 |
| Anxious temperament | −1.81 | 0.70 | 0.010 | 0.17 | 0.04–0.65 |
| YMRS score | 0.10 | 0.06 | 0.067 | 1.11 | 0.99–1.24 |
| Constant | −1.04 | 1.22 | 0.396 | 0.35 | |

Fit index of this model: $\chi^2$ = 25.84 (*p*-value < 0.05), Cox-Snell $R^2$ = 0.09, Hosmer–Lemeshow test *p* = 0.678, sensitivity = 0.32, specificity = 0.89, positive predictive value = 0.66, negative predictive value = 0.67, AUC of ROC = 0.66

dependent variable: diagnosis of MDD (1) and BD-II (2)

7 independent variables: scores of 5 subscales of the TEMPS-A and the severity of depressive and manic symptoms (PHQ-9 and YMRS scores, respectively)

BD-II, bipolar II disorder; MDD, major depressive disorder

**Table 4. Stepwise multivariate logistic regression analysis of the diagnosis of BD-II and BD-I.**

| Variable | Stepwise analysis | | | | |
|---|---|---|---|---|---|
| | B | S.E. | *p*-value | OR | 95%CI |
| Hyperthymic temperament | 2.01 | 0.82 | 0.014 | 7.44 | 1.50–36.94 |
| Constant | −3.12 | 1.02 | 0.002 | 0.04 | |

Fit index of this model: $\chi^2$ = 6.20 (*p*-value < 0.05), Cox-Snell $R^2$ = 0.04, Hosmer–Lemeshow test $p$ = 0.378, sensitivity = 0.16, specificity = 0.97, positive predictive value = 0.75, negative predictive value = 0.69, AUC of ROC = 0.57

dependent variable: the diagnosis of BD-II (1) and BD-I (2)

7 independent variables: scores of 5 subscales of the TEMPS-A and the severity of depressive and manic symptoms (PHQ-9 and YMRS scores, respectively)

BD- II, bipolar II disorder; BD- I, bipolar I disorder

that temperaments are more than just forme frustes of mood disorders and temperamental dysregulation is present in the subclinical stages, before patients experience mood episodes [25, 26]. Based on their descriptions and our present results, our findings may indicate that clinicians should keep in mind the possibility that MDD patients with high cyclothymic and low anxious scores on TEMPS-A may subsequently have manic/hypomanic episodes, and BD-II patients with a high hyperthymic temperament score may have manic episodes. We believe that this diagnostic conversion should be kept in mind clinically, although we should verify the role of affective temperaments prospectively.

The periods of the illness of the subjects were various. As shown in S4 Table, approximately 30% of the patients in each disease group were in remission. From a perspective of the impact of affective symptoms on the questionnaire answers [19], it is preferable that all subjects are in euthymic state when answering the questionnaires. However, we aimed to verify whether TEMPS-A was useful for the differential diagnosis of mood disorders in real-world clinical settings, and therefore, we included patients with various symptom severity. Moreover, we performed multivariate logistic regression analyses including the severity of mood symptoms and 5 affective temperaments as independent variables, considering the impact of them on TEMPS-A.

This is the first report to our knowledge that used multivariate analysis to show that affective temperament is a statistically significant differentiating factor of mood disorders, considering BD-I and BD-II separately, and taking into account manic and depressive symptoms as well as the 5 affective temperaments. On the other hand, Di Florio et al. compared patients with MDD and BD, taking into account depressive symptoms and hypomanic symptoms at the time of temperament assessment, but found no significant differences in all the subscales of the short version of the TEMPS-A [27]. A possible interpretation of this discrepancy is that, in the study by Di Florio et al., the MDD group included only the subjects diagnosed with recurrent MDD with hypomanic features, because recurrent major depressive episodes is a predictor of bipolarity [18]. The utilization of the short version of the TEMPS-A may also explain the difference between the results of their studies and ours, as the short version of the TEMPS-A (39 items) shows only weak or moderate correlation with the full version of the TEMPS-A (109/110 items) [28].

We conducted multivariate logistic regression analyses using the forced entry method as well as the backward stepwise selection method, and confirmed the robustness of the results of the original analysis of BD-I versus BD-II, although the fit index of this model might not be accepted. On the other hand, we obtained partially different results using the forced entry method from the original results. In the forced entry method, only cyclothymic temperament score was a significant differentiating factor of the diagnosis of BD-I and BD-II from the

diagnosis of MDD, and anxious temperament and YMRS scores were not significant differentiating factors of BD-I and BD-II versus MDD. The reason for this discrepancy might be that, although we attempted to construct the best model by taking into account mood symptoms and other affective temperaments, there is still a possibility that the effects of the excluded variables in the backward stepwise selection method were adjusted inadequately. We should hence confirm the reproducibility of our original results using larger samples in the future.

In addition, this study has several limitations. First, the design of our study was cross-sectional. Therefore, patients with MDD in this study may subsequently have a manic/hypomanic episode, leading to changes in their diagnoses. We should hence conduct follow-up observations to confirm their diagnoses. Secondly, all of the subjects included in the present study were patients treated in general or university hospitals, and consequently, there may be sampling biases.

In conclusion, the present study identified cyclothymic and anxious temperaments as differentiating factors of the diagnosis of BD compared with MDD, and hyperthymic temperament as a differentiating factor of BD-I compared with BD-II. These results indicate that evaluating affective temperaments with TEMPS-A in patients with mood disorders may be useful for distinguishing between MDD, BD-I, and BD-II.

## Supporting information

**S1 Table. Multivariate logistic regression analysis of the diagnosis of MDD and BD-I by the forced entry method.**
(DOCX)

**S2 Table. Multivariate logistic regression analysis of the diagnosis of MDD and BD-II by the forced entry method.**
(DOCX)

**S3 Table. Multivariate logistic regression analysis of the diagnosis of BD-II and BD-I using the forced entry method.**
(DOCX)

**S4 Table. Depressive symptoms assessed by the PHQ-9 and manic symptoms assessed by the YMRS in the subjects.**
(DOCX)

## Acknowledgments

We thank the patients who participated in this study.

## Author Contributions

**Conceptualization:** Chihiro Morishita, Rie Kameyama, Hiroyuki Toda, Takeshi Inoue.

**Data curation:** Rie Kameyama, Hiroyuki Toda, Takeshi Inoue.

**Formal analysis:** Chihiro Morishita.

**Investigation:** Rie Kameyama, Hiroyuki Toda.

**Methodology:** Chihiro Morishita.

**Project administration:** Chihiro Morishita, Jiro Masuya, Masahiko Ichiki, Ichiro Kusumi, Takeshi Inoue.

**Supervision:** Jiro Masuya, Masahiko Ichiki, Ichiro Kusumi, Takeshi Inoue.

**Validation:** Chihiro Morishita, Rie Kameyama, Hiroyuki Toda, Jiro Masuya, Masahiko Ichiki, Ichiro Kusumi, Takeshi Inoue.

**Writing – original draft:** Chihiro Morishita, Rie Kameyama, Hiroyuki Toda, Jiro Masuya, Masahiko Ichiki, Ichiro Kusumi, Takeshi Inoue.

**Writing – review & editing:** Chihiro Morishita, Rie Kameyama, Hiroyuki Toda, Jiro Masuya, Masahiko Ichiki, Ichiro Kusumi, Takeshi Inoue.

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
