## [Decision Letter · Decision Letter 0]

20 Feb 2020

PONE-D-20-02514

Utility of TEMPS-A in differentiation between major depressive disorder, bipolar I disorder, and bipolar II disorder

PLOS ONE

Dear Professor Inoue,

Thank you for submitting your manuscript to PLOS ONE. After careful consideration, we feel that it has merit but does not fully meet PLOS ONE’s publication criteria as it currently stands. Therefore, we invite you to submit a revised version of the manuscript that addresses the points raised during the review process.

The three reviewers addressed several major and minor concerns about your manuscript. Please revise your manuscript carefully.

We would appreciate receiving your revised manuscript by Apr 05 2020 11:59PM. To enhance the reproducibility of your results, we recommend that if applicable you deposit your laboratory protocols in protocols.io, where a protocol can be assigned its own identifier (DOI) such that it can be cited independently in the future. For instructions see: http://journals.plos.org/plosone/s/submission-guidelines#loc-laboratory-protocols

We look forward to receiving your revised manuscript.

Kind regards,

Kenji Hashimoto, PhD

Academic Editor

PLOS ONE

Journal Requirements:

"I have read the journal's policy and the authors of this manuscript have the following competing interests:

Jiro Masuya has received personal compensation from Otsuka Pharmaceutical, Eli Lilly, Astellas, and Meiji Yasuda Mental Health Foundation, and grants from Pfizer.

Masahiko Ichiki has received personal compensation from Otsuka Pharmaceutical, Pfizer, Eli Lilly, Mitsubishi Tanabe Pharma, Mochida Pharmaceutical, Meiji Seika Pharma, Janssen Pharmaceutical, Takeda Pharmaceutical, MSD, Dainippon Sumitomo Pharma, and Eisai; grants from Otsuka Pharmaceutical, Eli Lilly, Eisai, Shionogi, Takeda Pharmaceutical, MSD, and Pfizer; and is a member of the advisory board of Meiji Seika Pharma.

Ichiro Kusumi has received personal compensation from Astellas, Chugai Pharmaceutical, Daiichi Sankyo, Dainippon Sumitomo Pharma, Eisai, Eli Lilly, Janssen Pharmaceutical, Kyowa Hakko Kirin, Meiji Seika Pharma, MSD, Nippon Chemiphar, Novartis Pharma, Ono Pharmaceutical, Otsuka Pharmaceutical, Pfizer, Mitsubishi Tanabe Pharma, Shionogi, and Yoshitomiyakuhin; has received research/grant support from AbbVie GK, Asahi Kasei Pharma, Astellas, Boehringer Ingelheim, Chugai Pharmaceutical, Daiichi Sankyo, Dainippon Sumitomo Pharma, Eisai, Eli Lilly, GlaxoSmithKline, Kyowa Hakko Kirin, Meiji Seika Pharma, MSD, Novartis Pharma, Ono Pharmaceutical, Otsuka Pharmaceutical, Pfizer, Takeda Pharmaceutical, Tanabe Mitsubishi Pharma, Shionogi, and Yoshitomiyakuhin; and is a member of the advisory board of Dainippon Sumitomo Pharma and Tanabe Mitsubishi Pharma.

Takeshi Inoue has received personal compensation from Mochida Pharmaceutical, Takeda Pharmaceutical, Eli Lilly, Janssen Pharmaceutical, MSD, Taisho Toyama Pharmaceutical, Yoshitomiyakuhin, and Daiichi Sankyo; grants from Shionogi, Astellas, Tsumura, and Eisai; grants and personal fees from Otsuka Pharmaceutical, Dainippon Sumitomo Pharma, Mitsubishi Tanabe Pharma, Kyowa Pharmaceutical Industry, Pfizer, Novartis Pharma, and Meiji Seika Pharma; and is a member of the advisory boards of Pfizer, Novartis Pharma, and Mitsubishi Tanabe Pharma.

All other authors declare that they have no actual or potential conflicts of interest associated with this study."

Reviewers' comments:

Reviewer's Responses to Questions

**Comments to the Author**

1. Is the manuscript technically sound, and do the data support the conclusions?

Reviewer #1: No

Reviewer #2: Partly

Reviewer #3: Yes

2. Has the statistical analysis been performed appropriately and rigorously? 

Reviewer #1: No

Reviewer #2: Yes

Reviewer #3: Yes

3. Have the authors made all data underlying the findings in their manuscript fully available?

Reviewer #1: Yes

Reviewer #2: Yes

Reviewer #3: Yes

4. Is the manuscript presented in an intelligible fashion and written in standard English?

Reviewer #1: Yes

Reviewer #2: No

Reviewer #3: Yes

5. Review Comments to the Author

Reviewer #1: The authors investigated the utility of TEMPS-A for differential diagnosis of major depressive disorder (MDD) and bipolar disorders (BDs). They found that cyclothymic and anxious temperament scores were associated with the diagnoses of BDs versus MDD. They also found that hyperthymic temperament score was associated with the diagnoses of BD-I versus BD-II. The topic seems to be interesting for researchers in this field. However, I have the following concerns.

1. Categorical variables should be assessed by the Chi-square test or Fisher’s exact test rather than the Kruskal-Wallis test.

2. Sensitivity analysis is lacking. The stepwise multivariate logistic regression model automatically selects the best combination of predictive variables from the entered independent variables. In Table 2, the 3 independent variables, i.e. cyclothymic, anxious and YMRS, were selected for the best model, and the other variables, i.e. PHQ-9 and the other 3 TEMPS-A subscales, were just excluded from the model. This does not mean that the selected best model was adjusted for the effects of the excluded variables. The model in the Table 2 was not adjusted for the effects of PHQ-9 and the other 3 TEMPS-A subscales. The same is true for the results in Table 3 and 4. Therefore, in order to confirm the robustness of the original results, the authors should conduct the multivariate logistic regression analyses mandatorily including the 7 independent variables, and ideally demographic variables. These results from the sensitivity analyses should be mentioned in the main text and might be presented as supplementary tables.

3. In the first paragraph of the Discussion, “after adjusting for manic and depressive symptoms and 5 affective temperaments.” As mentioned above, the selected models did not adjust for several excluded variables.

4. It seems to be difficult for clinicians to predict the accurate differential diagnosis of MDD, BD-I and BD-II using TEMPS-A scores due to their relatively lower AUC values. This point should be briefly discussed. The authors need to tone down the language in the Conclusion, especially for predictive ability and accuracy.

5. In the Conclusion of the Abstract, the words “at early stages” should be removed.

Reviewer #2: This is a cross-sectional-study in the differentiation between major depressive disorder, bipolar I disorder, and bipolar II disorder using TEMPS-A. The study is interesting and important to patients with mood disorders. However, there are some concerns.

1. In introduction section, authors stated “Differentiating between major depressive disorder (MDD), bipolar I disorder (BD-I), and bipolar II disorder (BD-II) in the early stages of disease is clinically important.” Why is the differentiation needed? Authors need to explain in more detail.

2. In introduction section, authors stated “In recent years, the association between mood disorders and temperament characteristics has gathered much attention. ” What is the recent trend about these associations concretely?

3. In introduction section, there is no hypothesis. Authors need to clarify the hypotheses of this study, and discussion section should be made in accordance with authors’ hypotheses.

4. In methods section, authors described “TEMPS-A was measured at the time of visit.” What does this visit mean? First visit? Continuous visit?

5. In methods section, a statement as to whether your sample can be considered representative of a larger population is needed.

6. In methods section, authors need to clarify the sample size calculation.

7. Kawamura et al demonstrated that temperaments evaluated by TEMPS-A did not change substantially over 6 years (Kawamura Y, Akiyama T, Shimada T, Minato T, Umekage T, Noda Y, et al. Six-year stability of affective temperaments as measured by TEMPS-A. Psychopathology. 2010; 43(4):240–7. ) However, inpatients’ data in this study are over 8 years. How do authors explain about this point?

8. This study design is cross-sectional as authors mentioned in the lmitation section. So, the description, “predictor, predictive factor” are unsuitable.

9. The AUC between BD-I vs MDD is 0.708, between BD-II vs MDD is 0.655, between BD-I vs BD-II is 0.573. These AUCs are relatively low accuracy. I also think “predictor, predictive factor” is overstatement. Authors have to mention this point.

10. There are a few repetitions in the text. The paper could be improved by further editing.

Reviewer #3: The aim of this study is to evaluation of affective temperament measuring by TEMPS-A autoquestionaire in affective disorder and analyze of the utility of affective temperament dimensions in prediction of differential diagnosis. This study is interesting, however I have some remarks listed below:

1. There is a lack of detail about the data collection instrument. The PHQ-9 questionaire should be further described.

2. What is the reliability and validity of scales used?

3. The sample size was 346, 157 outpatients and 189 inpatients. How was it determined? Which is the method was used to recruit the sample? What are the eligibility criteria of the study sample?

4. The demographic data and the clinical characteristic should be present in method section (subject), not in results section.

4. Is the investigated group homogenous? What was the period of the illness (acute, partial remission) and the intensity of affective symptoms during the research procedure?

5. What was the current medication received by patients? Did authors observed differences in dosage of the drugs received by inpatients and outpatients?

4. In discussion - consider adding more recommendations for practice.

6. PLOS authors have the option to publish the peer review history of their article (what does this mean?). If published, this will include your full peer review and any attached files.

Reviewer #1: No

Reviewer #2: No

Reviewer #3: No

---

## [Author Response · Author response to Decision Letter 0]

4 Apr 2020

Point- by-point responses to the Reviewer’s comments

Reviewer #1

Comment 1: Categorical variables should be assessed by the Chi-square test or Fisher’s exact test rather than the Kruskal-Wallis test.

Response:

As you indicated, in general, differences between 2 groups are analyzed by the Chi-square test or Fisher’s exact test. However, in this study, we used the Kruskal-Wallis test to analyze differences among 3 groups, and planned to use the Bonferroni test for multiple comparison between groups if there were statistically significant differences among the 3 groups. This is because a multiple comparison cannot be performed using the Chi-square test or the Fisher’s exact test.

Comment 2: Sensitivity analysis is lacking. The stepwise multivariate logistic regression model automatically selects the best combination of predictive variables from the entered independent variables. In Table 2, the 3 independent variables, i.e. cyclothymic, anxious and YMRS, were selected for the best model, and the other variables, i.e. PHQ-9 and the other 3 TEMPS-A subscales, were just excluded from the model. This does not mean that the selected best model was adjusted for the effects of the excluded variables. The model in the Table 2 was not adjusted for the effects of PHQ-9 and the other 3 TEMPS-A subscales. The same is true for the results in Table 3 and 4. Therefore, in order to confirm the robustness of the original results, the authors should conduct the multivariate logistic regression analyses mandatorily including the 7 independent variables, and ideally demographic variables. These results from the sensitivity analyses should be mentioned in the main text and might be presented as supplementary tables.

Response:

Thank you for pointing out this important point. In accordance with the comment, we conducted multivariate logistic regression analyses using the forced entry method including the 7 independent variables, although we could not include demographic variables owing to the sample size. We have added these results as Tables S1-3, and a description of the results was added to the Results and Discussion sections of the revised manuscript, as follows.

In the Results section (page 19, lines 251-257)

“We also conducted the multivariate logistic regression analyses using the forced entry method, including PHQ-9 and YMRS scores, and TEMPS-A subscores as independent variables, to confirm the robustness of our results. The results are shown in Tables S1-3. Tables S1 and S2 show that only cyclothymic temperament score is a significant differentiating factor of the diagnosis of BD-I and BD-II from the diagnosis of MDD. Table S3 demonstrates that hyperthymic temperament score is a specific factor for the differential diagnosis of BD-I versus the diagnosis of BD-II.”

In the Discussion section (page 22, lines 297-309)

“We conducted multivariate logistic regression analyses using the forced entry method as well as the backward stepwise selection method, and confirmed the robustness of the results of the original analysis of BD-I versus BD-II, although the fit index of this model might not be accepted. On the other hand, we obtained partially different results using the forced entry method from the original results. In the forced entry method, only cyclothymic temperament score was a significant differentiating factor of the diagnosis of BD-I and BD-II from the diagnosis of MDD, and anxious temperament and YMRS scores were not significant differentiating factors of BD-I and BD-II versus MDD. The reason for this discrepancy might be that, although we attempted to construct the best model by taking into account mood symptoms and other affective temperaments, there is still a possibility that the effects of the excluded variables in the backward stepwise selection method were adjusted inadequately. We should hence confirm the reproducibility of our original results using larger samples in the future.”

Comment 3: In the first paragraph of the Discussion, “after adjusting for manic and depressive symptoms and 5 affective temperaments.” As mentioned above, the selected models did not adjust for several excluded variables.

Response:

We thank you for pointing this out. As you suggested, there is a possibility that we inadequately adjusted for several excluded variables in our original analyses. We hence changed the phrase “after adjusting for manic and depressive symptoms and 5 affective temperaments” to “taking into account manic and depressive symptoms as well as 5 affective temperaments” in the revised manuscript.

Comment 4: It seems to be difficult for clinicians to predict the accurate differential diagnosis of MDD, BD-I and BD-II using TEMPS-A scores due to their relatively lower AUC values. This point should be briefly discussed. The authors need to tone down the language in the Conclusion, especially for predictive ability and accuracy.

Response:

We agree that it appears to be difficult for clinicians to predict an accurate differential diagnosis of MDD, BD-I, and BD-II by only TEMPS-A subscores, owing to their relatively lower AUC values. We hence changed the term “predictor” to “differentiating factor,” and deleted the term “accurately” in the Results and Discussion sections of the revised manuscript. We also toned down our conclusions.

Comment 5: In the Conclusion of the Abstract, the words “at early stages” should be removed.

Response:

Thank you for your suggestion. We removed the words “at early stages” from the revised manuscript.

Reviewer #2

Comment 1: In introduction section, authors stated “Differentiating between major depressive disorder (MDD), bipolar I disorder (BD-I), and bipolar II disorder (BD-II) in the early stages of disease is clinically important.” Why is the differentiation needed? Authors need to explain in more detail.

Response:

Thank you for your suggestion. To make this point clear, we added the reasons to the Introduction section, as follows (page 5, lines 52-59).

“Differentiating between major depressive disorder (MDD), bipolar I disorder (BD-I), and bipolar II disorder (BD-II) in the early stages of disease is clinically important (Altamura et al., 2010; Drancourt et al., 2013), because clinicians should take different treatment approaches for the 3 disorders and inappropriate treatment can be associated with poor prognoses. For example, clinicians should not provide antidepressant monotherapy to patients with BD, particularly those with BD-I, according to many treatment guidelines for mood disorders ([3] Kanba et al., 2013; [4] Yatham et al., 2018), and inappropriate treatment of BD and an extended duration of untreated BD may increase the risk of mood instabilities and suicide attempts (Altamura et al., 2010; Drancourt et al., 2013).”

Furthermore, the following 2 references were added to the References section.

[3] Kanba S, Kato T, Terao T, Yamada K, Committee for Treatment Guidelines of Mood Disorders JSoMD. Guideline for treatment of bipolar disorder by the Japanese Society of Mood Disorders, 2012. Psychiatry Clin Neurosci. 2013;67(5):285-300. https://doi.org/10.1111/pcn.12060. PMID: 23773266.

[4] Yatham LN, Kennedy SH, Parikh SV, Schaffer A, Bond DJ, Frey BN, et al. Canadian Network for Mood and Anxiety Treatments (CANMAT) and International Society for Bipolar Disorders (ISBD) 2018 guidelines for the management of patients with bipolar disorder. Bipolar Disord. 2018;20(2):97-170. https://doi.org/10.1111/bdi.12609. PMID: 29536616; PubMed Central PMCID: PMCPMC5947163.

Comment 2: In introduction section, authors stated “In recent years, the association between mood disorders and temperament characteristics has gathered much attention. ” What is the recent trend about these associations concretely?

Response:

Thank you for pointing out this important point. As mentioned in our original manuscript, several studies have shown the possibility of the usefulness of TEMPS-A to differentiate between MDD and BD. Moreover, in accordance with the comment, we added the following description to the Introduction section (page 6, lines 67-73).

“Vazquez et al. showed the association between some affective temperaments on TEMPS-A and a suicidal risk in both psychiatric and general population samples ([8] Vazquez et al., 2018). The association between some affective temperaments on TEMPS-A and treatment resistance in MDD and BD patients has also been studied (Toda et al., 2015; Toda et al., 2018; Takeshima and Oka, 2016). Moreover, Goto et al. suggested that cyclothymic and hyperthymic temperaments were associated with bipolarity, and also investigated the association between treatments and remission rates in patients with bipolarity ([12] Goto et al., 2011).”

Furthermore, the following 2 references were added to the References section.

[8] Vazquez GH, Gonda X, Lolich M, Tondo L, Baldessarini RJ. Suicidal risk and affective temperaments, evaluated with the TEMPS-A scale: A systematic review. Harv Rev Psychiatry. 2018;26(1):8-18. https://doi.org/10.1097/HRP.0000000000000153. PMID: 29303918.

[12] Goto S, Terao T, Hoaki N, Wang Y. Cyclothymic and hyperthymic temperaments may predict bipolarity in major depressive disorder: a supportive evidence for bipolar II1/2 and IV. J Affect Disord. 2011;129(1-3):34-8. https://doi.org/10.1016/j.jad.2010.07.016. PMID: 20699193.

Comment 3: In introduction section, there is no hypothesis. Authors need to clarify the hypotheses of this study, and discussion section should be made in accordance with authors’ hypotheses.

Response:

We apologize that there was no hypothesis in the original manuscript. We included our hypothesis that each subscale of affective temperaments assessed by TEMPS-A is a differentiating factor of the diagnosis of MDD, BD-I, and BD-II in the Introduction section of the revised manuscript (page 8, lines 103-104).

Comment 4: In methods section, authors described “TEMPS-A was measured at the time of visit.” What does this visit mean? First visit? Continuous visit?

Response:

Thank you for your valuable suggestion. We rephrased our expression in the Methods section, as follows (page 12, lines 154-157).

“The inpatients filled out the TEMPS-A at the time of hospital discharge, after their symptoms were improved by adequate treatment. The outpatients filled out the TEMPS-A at some point in their continuous visits, but not at their first visit. In other words, they completed it after they received treatment for months or years.”

Comment 5: In methods section, a statement as to whether your sample can be considered representative of a larger population is needed.

Response:

Thank you for your valuable suggestion. We believe that our sample may not be a nationally representative sample for various reasons. To make this point clear, we added the following sentences to the Methods section (page 10, lines132-136).

“The sample might not be a nationally representative sample, as the sample was limited to a convenience sample, and consisted of only patients who received treatment at general or university hospitals in specific parts of Japan. In other words, the sample did not include patients who received treatment at psychiatric hospitals or clinics.”

Comment 6: In methods section, authors need to clarify the sample size calculation.

Response:

Thank you for your suggestion. Approximately 70 or more subjects for each disorder group were required for our analyses, because we planned to conduct multivariate logistic regression analyses including 7 factors as independent variables. This description was added to the Methods section of the revised manuscript (page 9, lines 113-115).

Comment 7: Kawamura et al demonstrated that temperaments evaluated by TEMPS-A did not change substantially over 6 years (Kawamura Y, Akiyama T, Shimada T, Minato T, Umekage T, Noda Y, et al. Six-year stability of affective temperaments as measured by TEMPS-A. Psychopathology. 2010; 43(4):240–7.) However, inpatients’ data in this study are over 8 years. How do authors explain about this point?

Response:

As you pointed out, Kawamura et al. showed the 6-year stability of affective temperaments measured by TEMPS-A in a nonclinical adult population. However, our design was cross-sectional, and we did not follow up the patients prospectively. Moreover, to avoid confusion, we changed the expression of subjects from “those who received treatment from April 2012 to April 2013” to “those who received treatment between April 2012 and April 2013,” and from “those who received treatment from January 2010 to December 2017” to “those who received treatment between January 2010 and December 2017” in the revised manuscript.

Comment 8: This study design is cross-sectional as authors mentioned in the lmitation section. So, the description, “predictor, predictive factor” are unsuitable.

Response:

As you pointed out, we also think that TEMPS-A subscores are not necessarily predictive factors, owing to the study design. Therefore, we replaced the description “predictor and predictive factor” with the phrase “differentiating factor” throughout the revised manuscript.

Comment 9: The AUC between BD-I vs MDD is 0.708, between BD-II vs MDD is 0.655, between BD-I vs BD-II is 0.573. These AUCs are relatively low accuracy. I also think “predictor, predictive factor” is overstatement. Authors have to mention this point.

Response:

We apologize for using this overstatement. We also think that it is difficult for clinicians to predict the diagnosis of MDD, BD-I, and BD-II by only TEMPS-A, owing to their relatively lower AUC values. As mentioned in our answer to comment 8, we toned down our statement in the revised manuscript. However, we believe that subscores on the TEMPS-A may play a supplementary role in the differential diagnoses of mood disorders. In other words, utilizing TEMPS-A with other factors may help clinicians to differentiate between the diagnosis of MDD, BD-I, and BD-II.

Comment 10: There are a few repetitions in the text. The paper could be improved by further editing.

Response:

We apologize for this point. We reread the manuscript and deleted the repetitions.

Reviewer #3

Comment 1: There is a lack of detail about the data collection instrument. The PHQ-9 questionaire should be further described.

Response:

We apologize for our insufficient explanation. We added the following explanation to the Methods section of the revised manuscript (page 12, lines 159-162).

“The PHQ-9 is a self-reported questionnaire consisting of 9 items on a 4-point Likert scale, used as an index of depressive symptom severity and as a screening test for major depressive episodes. The Japanese version of the PHQ-9 was used in this study.”

Comment 2: What is the reliability and validity of scales used?

Response:

Thank you for pointing this out. To make this point clear, we added an explanation together with 2 references of the scales that were used, to the Methods section, as follows.

“The validity and reliability of TEMPS-A for psychiatric disorders, particularly for mood disorders, was suggested by Akiskal et al. ([7] Akiskal et al., 2005), and the validity and reliability of the Japanese version was also shown by Matsumoto et al. (Matsumoto et al., 2005).” (page 12, lines 148-151)

“Some studies suggested that the PHQ-9 yields an index of depressive symptom severity and has diagnostic validity, and also validates the Japanese version of the PHQ-9 (Muramatsu et al., 2007; Spitzer et al., 1999; [23] Muramatsu et al., 2018).” (page 12, line 162 to page 13, line 164)

“The construct validity and reliability of YMRS was demonstrated by Young et al. (Young et al., 1978).” (page 13, lines 168-169)

Furthermore, the following 2 references were added to the References section.

[7] Akiskal HS, Akiskal KK, Haykal RF, Manning JS, Connor PD. TEMPS-A: progress towards validation of a self-rated clinical version of the Temperament Evaluation of the Memphis, Pisa, Paris, and San Diego Autoquestionnaire. J Affect Disord. 2005;85(1-2):3-16. https://doi.org/10.1016/j.jad.2004.12.001. PMID: 15780671.

[23] Muramatsu K, Miyaoka H, Kamijima K, Muramatsu Y, Tanaka Y, Hosaka M, et al. Performance of the Japanese version of the Patient Health Questionnaire-9 (J-PHQ-9) for depression in primary care. Gen Hosp Psychiatry. 2018;52:64-9. https://doi.org/10.1016/j.genhosppsych.2018.03.007. PMID: 29698880.

Comment 3: The sample size was 346, 157 outpatients and 189 inpatients. How was it determined? Which is the method was used to recruit the sample? What are the eligibility criteria of the study sample?

Response:

We apologize that we did not adequately explain about our sample. As mentioned in our responses to the Comments 5 and 6 by Reviewer #2 partly, we added the following explanations to the Methods section.

“Approximately 70 or more subjects for each disorder group were required for our analyses, because we planned to conduct multivariate logistic regression analyses including 7 factors as independent variables.” (page 9, lines 113-115)

“The inclusion criteria of the patients were as follows: (1) a principal diagnosis of MDD, BD-I, or BD-II according to the DSM-Ⅳ-TR criteria; (2) 20 years of age or older; (3) the ability to complete the self-reported questionnaires; and (4) the ability to provide written informed consent. The exclusion criteria were as follows: (1) having serious physical or mental symptoms that hinder the completion of the self-reported questionnaires; (2) having organic mental disorders or a previous history of them; (3) meeting the diagnostic criteria of substance-use disorders; and (4) having a diagnosis of axis II according to the DSM-Ⅳ-TR criteria. Patients meeting the eligibility criteria were informed about our research by their doctors in charge, and those who gave written consent were included in the study. The sample might not be a nationally representative sample, as the sample was limited to a convenience sample, and consisted of only patients who received treatment at general or university hospitals in specific parts of Japan. In other words, the sample did not include patients who received treatment at psychiatric hospitals or clinics.” (page 9, line 123 to page 10, line 136)

Comment 4: The demographic data and the clinical characteristic should be present in method section (subject), not in results section.

Response:

In accordance with this comment, we moved the demographic data and the clinical characteristics (Table 1) to the Methods section of our revised manuscript (page10, line140 to page 11, line 144). The statistical results of these data were shown in the Results section.

Comment 5: Is the investigated group homogenous? What was the period of the illness (acute, partial remission) and the intensity of affective symptoms during the research procedure?

Response:

Thank you for your suggestion. The investigated group was not homogenous. To explain the investigated group more clearly, we added the following description to the Discussion section (page20, line 275 to page 21, line 283) and Table S4 of the revised manuscript.

“The periods of the illness of the subjects were various. As shown in Table S4, approximately 30% of the patients in each disease group were in remission. From a perspective of the impact of affective symptoms on the questionnaire answers (Baba et al., 2014), it is preferable that all subjects are in euthymic state when answering the questionnaires. However, we aimed to verify whether TEMPS-A was useful for the differential diagnosis of mood disorders in real-world clinical settings, and therefore, we included patients with various symptom severity. Moreover, we performed multivariate logistic regression analyses including the severity of mood symptoms and 5 affective temperaments as independent variables, considering the impact of them on TEMPS-A.”

Comment 6: What was the current medication received by patients? Did authors observed differences in dosage of the drugs received by inpatients and outpatients?

Response:

Most patients used several types of psychotropics, but we did not investigate their use of medication in this study. We would hence like to investigate this point in the future.

Comment 7: In discussion - consider adding more recommendations for practice.

Response:

In accordance with the suggestion, we added more recommendations, with 2 references to the Discussion section, as follows (page 20, lines 266-274).

“Affective temperament is a concept proposed by Akiskal and his colleagues. They described that temperaments are more than just forme frustes of mood disorders and temperamental dysregulation is present in the subclinical stages, before patients experience mood episodes ([25] Akiskal and Mallya, 1987; [26] Akiskal et al., 1995). Based on their descriptions and our present results, our findings may indicate that clinicians should keep in mind the possibility that MDD patients with high cyclothymic and low anxious scores on TEMPS-A may subsequently have manic/hypomanic episodes, and BD-II patients with a high hyperthymic temperament score may have manic episodes. We believe that this diagnostic conversion should be kept in mind clinically, although we should verify the role of affective temperaments prospectively.”

Furthermore, the following references were added to the References section.

[25] Akiskal HS, Mallya G. Criteria for the "soft" bipolar spectrum: treatment implications. Psychopharmacol Bull. 1987;23(1):68-73. PMID: 3602332.

[26] Akiskal HS, Maser JD, Zeller PJ, Endicott J, Coryell W, Keller M, et al. Switching from 'unipolar' to bipolar II. An 11-year prospective study of clinical and temperamental predictors in 559 patients. Arch Gen Psychiatry. 1995;52(2):114-23. https://doi.org/10.1001/archpsyc.1995.03950140032004. PMID: 7848047.

---

## [Decision Letter · Decision Letter 1]

16 Apr 2020

Utility of TEMPS-A in differentiation between major depressive disorder, bipolar I disorder, and bipolar II disorder

PONE-D-20-02514R1

Dear Dr. Inoue,

We are pleased to inform you that your manuscript has been judged scientifically suitable for publication and will be formally accepted for publication once it complies with all outstanding technical requirements.

With kind regards,

Kenji Hashimoto, PhD

Section Editor

PLOS ONE

Additional Editor Comments (optional):

Reviewers' comments:

Reviewer's Responses to Questions

**Comments to the Author**

1. If the authors have adequately addressed your comments raised in a previous round of review and you feel that this manuscript is now acceptable for publication, you may indicate that here to bypass the “Comments to the Author” section, enter your conflict of interest statement in the “Confidential to Editor” section, and submit your "Accept" recommendation.

Reviewer #1: All comments have been addressed

Reviewer #2: All comments have been addressed

Reviewer #3: All comments have been addressed

2. Is the manuscript technically sound, and do the data support the conclusions?

Reviewer #1: Yes

Reviewer #2: Yes

Reviewer #3: Yes

3. Has the statistical analysis been performed appropriately and rigorously? 

Reviewer #1: Yes

Reviewer #2: Yes

Reviewer #3: Yes

4. Have the authors made all data underlying the findings in their manuscript fully available?

Reviewer #1: Yes

Reviewer #2: Yes

Reviewer #3: Yes

5. Is the manuscript presented in an intelligible fashion and written in standard English?

Reviewer #1: Yes

Reviewer #2: Yes

Reviewer #3: Yes

6. Review Comments to the Author

Reviewer #1: (No Response)

Reviewer #2: (No Response)

Reviewer #3: The article was corrected in line with the reviewer's comments. The study group is not homogeneous, which is a serious limitation of the study, however the authors characterized the study group much better than before. After revision, the article meets the publication criteria in PlosOne.

7. PLOS authors have the option to publish the peer review history of their article (what does this mean?). If published, this will include your full peer review and any attached files.

Reviewer #1: No

Reviewer #2: No

Reviewer #3: No

---

## [Editor Report · Acceptance letter]

13 May 2020

PONE-D-20-02514R1 

Utility of TEMPS-A in differentiation between major depressive disorder, bipolar I disorder, and bipolar II disorder 

Dear Dr. Inoue:

I am pleased to inform you that your manuscript has been deemed suitable for publication in PLOS ONE. Congratulations! Your manuscript is now with our production department. 

With kind regards,

on behalf of

Prof. Kenji Hashimoto 

Section Editor

PLOS ONE